# Measurement of Double-Differential Cross-Sections for Mesonless Charged Current Neutrino Scattering on Argon with MicroBooNE †

**Julia Book** 🆔 **on behalf of the MicroBooNE Collaboration**

Laboratory for Particle Physics and Cosmology (LPPC), Harvard University, Cambridge, MA 02138, USA;
jbook@g.harvard.edu
† Presented at the 23rd International Workshop on Neutrinos from Accelerators, Salt Lake City, UT, USA,
30–31 July 2022.

**Abstract:** The MicroBooNE liquid argon time projection chamber experiment is pursuing a broad range of neutrino physics measurements, including some of the first high-statistics results for neutrino–argon scattering cross-sections. At the neutrino energies relevant for MicroBooNE and its companion experiments in the Fermilab Short-Baseline Neutrino program, the dominant event topology involves mesonless final states containing one or more protons. A complete description of these events requires modeling the contributions of quasielastic and two-particle, two-hole neutrino interactions, as well as more inelastic reaction modes in which final state pions are reabsorbed by the residual nucleus. Refinements to the current understanding of these processes, informed by new neutrino cross-section data, will enable a precise and reliable interpretation of future measurements of neutrino oscillations and searches for exotic physics processes involving neutrinos. This proceeding presents the first double-differential cross-section results from MicroBooNE for mesonless charged current scattering of muon neutrinos on argon.

**Keywords:** LArTPC; neutrino; cross-section





## 1. Introduction

With the proliferation of liquid argon technology and planned large-scale liquid argon time projection chamber (LArTPC) experiments, a thorough understanding of the neutrino–argon cross-section holds a pivotal place in the advancement of the field of neutrino physics. As a mature experiment, MicroBooNEs contributions to cross-section measurements have great potential to improve understanding across the field, reducing cross-section and nuclear model uncertainties (one of the main sources of uncertainty for high-precision experiments) by up to a factor of five.

MicroBooNE is an 85 tonne LArTPC situated in the Booster Neutrino Beam (BNB) at Fermilab at a 469 m baseline [1]. The BNB produces neutrinos primarily at energies from 0.5 to 2.0 GeV. At these energies, quasielastic (QE) interactions dominate, as shown in Figure 1. Charged current quasielastic (CCQE) scattering events, where the neutrino scatters off of the argon nucleus creating a charged lepton, are the most common event topology in MicroBooNE, and CCQE events will continue to contribute significantly to long-baseline experiments.

Presented in this proceeding is the single-differential $CC0\pi Np$ event rate, along with the $CC0\pi 1p$ double-differential cross-section in terms of the transverse variables $\delta_{\alpha_T}$ and $\delta_{p_T}$ (defined in Section 3). The results in this proceeding were obtained using MicroBooNEs first three data-taking runs, with a total exposure of $6.79 \times 10^{20}$ protons on target. This proceeding presents the results described in detail in the following papers and public notes: [2,3] (the $CC0\pi 1p$ results) and [4] (the $CC0\pi Np$ results).

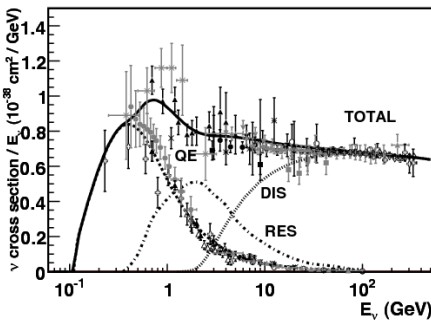

**Figure 1.** Neutrino cross-section as a function of energy [5].

## 2. Signal Definition and Event Selection

This proceeding presents results for two event categories: muon-neutrino events with no pions and any number of protons in the final state ($CC0\pi Np$), and muon-neutrino events with no pions and one proton in the final state ($CC0\pi 1p$). For the first category, $CC0\pi Np$ events, a singe-differential measurement was performed. For the second category, $CC0\pi 1p$ events, a double-differential measurement in terms of transverse variable measurement was performed.

The two measurements have similar energy requirements. For the $CC0\pi 1p$ measurement, we require the final state muon to have a reconstructed momentum $0.1 < p_\mu < 1.2$ GeV/c and the final state proton to have $0.3 < p_p < 1$ GeV/c. For the $CC0\pi Np$ measurement, we require the momentum of the outgoing muon to be greater than 0.10 GeV/c, and the momentum of the leading proton to have $0.25 < p_p < 1.2$ GeV/c.

In both selections, we further require that there are no pions (of any momentum) in the final state. To pass the selection, events must pass a series of quality cuts based on the signal definition and protons must be fully contained within the fiducial volume of the detector. For the $CC0\pi 1p$ measurement, muons must also be contained.

A thorough discussion of the event selection used in each of these measurements can be found in [3] for the $CC0\pi 1p$ measurement and [4] for the $CC0\pi Np$ measurement. The $CC0\pi 1p$ sample has a purity of $\approx$70% and an efficiency of $\approx$10% [3], while the $CC0\pi Np$ sample has a purity of $\approx$77% and an overall efficiency of $\approx$36% [4].

## 3. Single Transverse Variables

Neutrino energy is calculated from the measured kinetic energies of particles emitted by the neutrino interaction. The kinematic properties of these final state particles are affected by initial state effects and nuclear effects, including final state interactions (FSIs). Transverse variables are built to characterize these nuclear effects by quantifying the transverse momentum and angular motion of the muons and leading protons, as seen in Figure 2. The magnitude ($\delta p_T$) and angles ($\delta\alpha_T$, $\delta\phi_T$) of the leading proton's transverse momentum are shown, as well as the outgoing muon momentum $p_\mu$.

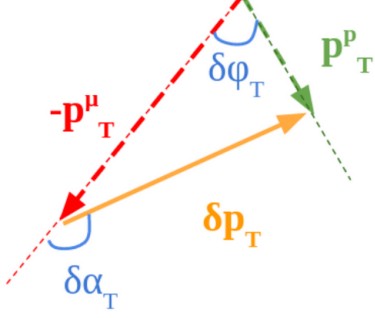

**Figure 2.** Schematic illustration of the transverse variables used in this work [3]. The neutrino travels in the z direction, where the x–y plane is the plane of the page.

## 4. Results

Selected results from the $CC0\pi1p$ [3] and $CC0\pi Np$ [4] cross-section studies are presented here.

Shown in Figure 3 is the single-differential $CC0\pi Np$ event rate compared to the GENIE (G18) [6] prediction. The $CC0\pi1p$ double-differential cross-section in terms of the trasnverse variables $\delta_{\alpha T}$ and $\delta_{p_T}$ is shown in Figure 4, along with predictions from GENIE and GiBUU [7] with and without FSI and the reduced $\chi^2$ for each. The dominant sources of systematic uncertainty in both measurements are the detector response, the flux, and cross-section uncertainties, with the detector response the largest source of uncertainty in the $CC0\pi Np$ measurements and flux and cross-section systematics the largest in the $CC0\pi1p$ measurements.

In this regard, the transverse variable cross-sections (Figure 3), strongly disfavoring the no FSI hypothesis, start pointing to areas where modeling improvements are needed. The transverse variable cross-sections (Figure 4) begin this work, strongly disfavoring the no FSI hypothesis and beginning to point to areas where modeling improvements are needed.

These results point to data MC disagreement and indicate that FSIs are present and dominate in specific parts of the phase space (as in Figure 4c), paving the way for nuclear interaction model discrimination that can be used in future measurements.

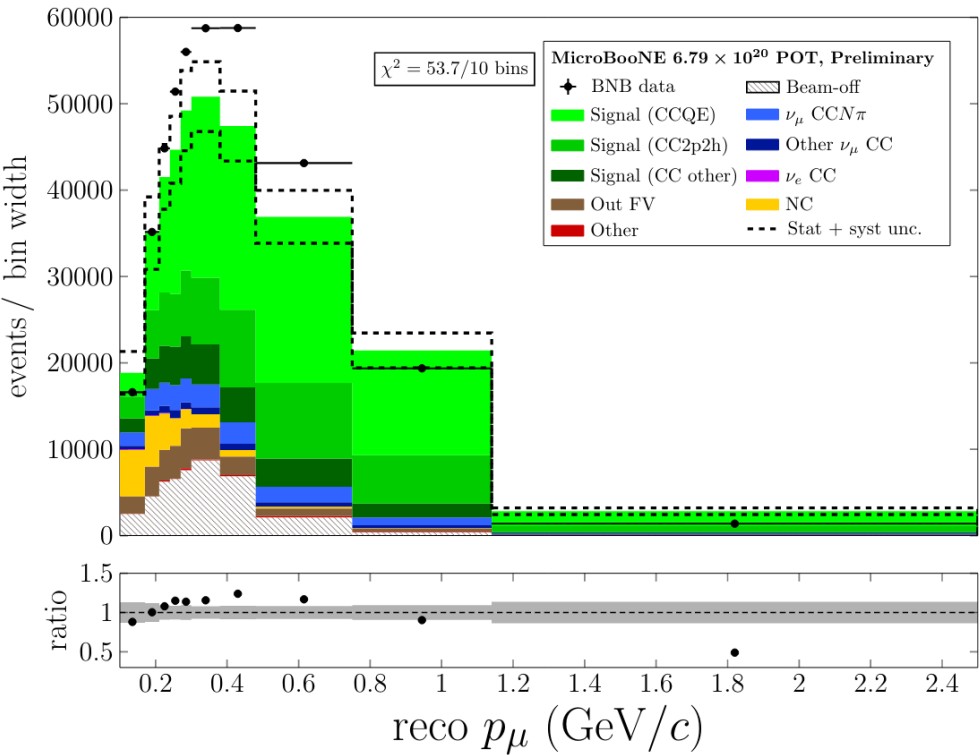

**Figure 3.** The single-differential $CC0\pi Np$ event rate in terms of the reconstructed muon momentum $p_\mu$ [4]. This event rate is obtained by integrating the 2D event distribution (in terms of $p_\mu, \cos\theta_\mu$) over all angles. The dashed lines indicate the total uncertainty in the GENIE Monte Carlo prediction.

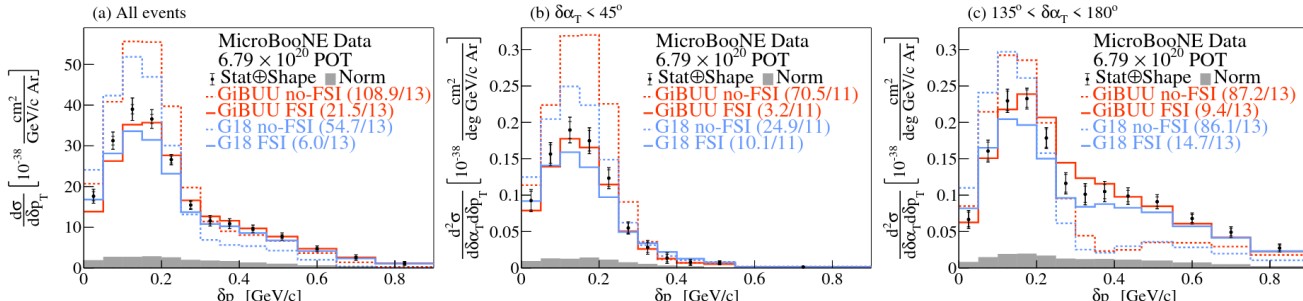

**Figure 4.** $CC1p0\pi$ cross-sections as a function of (**a**) $\delta_{p_T}$ for all events, and separated by (**b**) low (less than 45 degrees) and (**c**) high (above 135 degrees) $\delta\alpha_T$. We thereby present (**a**) single- and (**b**,**c**) double-differential cross-section measurements. The black points indicate data, while the gray bands show normalized systematic uncertainty. The inner error bars show statistical uncertainty only, while the outer error bars show the combined total (statistical and shape) uncertainty. The colored lines compare the predictions of GiBUU (orange) and GENIE (G18, blue) event generators with and without FSIs [2].

**Funding:** This research was performed by the MicroBooNE collaboration, supported by Fermilab, US Department of Energy, and the US National Science Foundation.

**Conflicts of Interest:** The author declare no conflicts of interest.

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
