# Peer review of "Measurement of Double-Differential Cross-Sections for Mesonless Charged Current Neutrino Scattering on Argon with MicroBooNEâ€"

_psf, doi:10.3390/psf8010033_

Round 1
Reviewer 1 Report
I suggest the following:
1) line90: Please provide a weblink or cite the paper also for reference nb [5] as done for the others
2) line 25: "scatters of" should be "neutrino scatters off the argon nucleus"
3) line36: instead of "two events selections", "two event categories"
4) line36-40: it is confusing, it sould be rephrased. In the introduction it is stated that the results presented refer to single-differential event rate for CC0piNp and double-differential cross-section for CC0pi1p events
5)line56-57: Was it supposed to be "The kinematic properties of these final state particles are affected by nuclear and initial state effects including final state interactions (FSI)"? Probably better: "The kinematic properties of these final state particles are affected by nuclear initial state effects, including also final state interactions (FSI)"
6)line60: "angles" instead of "angle"
7)line67: "transverse variables delta ALPHA T", the alpha is missing
8)line74-76: It should be rephrased, I suggest for example "In this regard, the transverse variable cross sections (figure 4), strongly disfavoring the no FSI hypothesis, start pointing to areas where modeling improvements are needed"
Reviewer 2 Report
The proceedings are well-written and I support publication in the present form
Author Response
No revisions required. Thank you for your feedback!